# Myocardial and Arrhythmic Spectrum of Neuromuscular Disorders in Children

**DOI:** 10.3390/biom11111578

**Published:** 2021-10-25

**Authors:** Anwar Baban, Valentina Lodato, Giovanni Parlapiano, Corrado di Mambro, Rachele Adorisio, Enrico Silvio Bertini, Carlo Dionisi-Vici, Fabrizio Drago, Diego Martinelli

**Affiliations:** 1The European Reference Network for Rare, Low Prevalence and Complex Diseases of the Heart-ERN GUARD-Heart, Department of Pediatric Cardiology and Cardiac Surgery, Heart and Lung Transplantation, Bambino Gesù Children Hospital and Research Institute, IRCCS, 00165 Rome, Italy; valentina.lodato@opbg.net (V.L.); corrado.dimambro@opbg.net (C.d.M.); rachele.adorisio@opbg.net (R.A.); fabrizio.drago@opbg.net (F.D.); 2Laboratory of Medical Genetics, Translational Cytogenomics Research Unit, Bambino Gesù Children Hospital and Research Institute, IRCCS, 00165 Rome, Italy; giovanni.parlapiano@opbg.net; 3The European Reference Network for Neuromuscular Disorders (ERN NMD), Unit of Neuromuscular and Neurodegenerative Disorders, Bambino Gesù Children Hospital and Research Institute, IRCCS, 00146 Rome, Italy; enricosilvio.bertini@opbg.net; 4Division of Metabolism, Bambino Gesù Children Hospital and Research Institute, IRCCS, 00165 Rome, Italy; carlo.dionisivici@opbg.net (C.D.-V.); diego.martinelli@opbg.net (D.M.)

**Keywords:** neuromuscular disorders, cardiac involvement, heterogeneity, age related penetrance, anticipation, rarity

## Abstract

Neuromuscular disorders (NMDs) are highly heterogenous from both an etiological and clinical point of view. Their signs and symptoms are often multisystemic, with frequent cardiac involvement. In fact, childhood onset forms can predispose a person to various progressive cardiac abnormalities including cardiomyopathies (CMPs), valvulopathies, atrioventricular conduction defects (AVCD), supraventricular tachycardia (SVT) and ventricular arrhythmias (VA). In this review, we selected and described five specific NMDs: Friedreich’s Ataxia (FRDA), congenital and childhood forms of Myotonic Dystrophy type 1 (DM1), Kearns Sayre Syndrome (KSS), Ryanodine receptor type 1-related myopathies (*RYR1*-RM) and Laminopathies. These changes are widely investigated in adults but less researched in children. We focused on these specific topics due their relative frequency and their potential unexpected cardiac manifestations in children. Moreover these conditions present different inheritance patterns and mechanisms of action. We decided not to discuss Duchenne and Becker muscular dystrophies due to extensive work regarding the cardiac aspects in children. For each described NMD, we focused on the possible cardiac manifestations such as different types of CMPs (dilated-DCM, hypertrophic-HCM, restrictive-RCM or left ventricular non compaction-LVNC), structural heart abnormalities (including valvulopathies), and progressive heart rhythm changes (AVCD, SVT, VA). We describe the current management strategies for these conditions. We underline the importance, especially for children, of a serial multidisciplinary personalized approach and the need for periodic surveillance by a dedicated heart team. This is largely due to the fact that in children, the diagnosis of certain NMDs might be overlooked and the cardiac aspect can provide signs of their presence even prior to overt neurological diagnosis.

## 1. Introduction

The genotyping of neuromuscular disorders (NMDs) is increasingly used as a clinical practice, even in young-aged individuals. Different underlying mechanisms including the cytoskeleton, the nuclear membrane, and mitochondrial integrity are well reported in NMDs [1]. In childhood onset NMDs, atypical or “isolated” primary cardiac manifestations are less frequent but are not rare [2,3]. This group may be overlooked, underestimated and inappropriately managed in clinical practice. The most frequently reported NMDs are the Duchenne and Becker muscular dystrophies. We decided not to discuss them in this review due to our extensive discussion concerning cardiac management in children [4]. Other conditions that more frequently present in children with peculiar phenocopies include Friedreich’s Ataxia, myotonic dystrophy (congenital form), Kearns Sayre, *RYR1* associated defects, and Laminopathies. These groups have a relatively higher frequency in children compared to other NMDs. In this review we have chosen to discuss them due to the increased cardiac and therapeutic options related to these disorders. Cardiac manifestations include: cardiomyopathies (CMPs) mainly in their dilated and hypertrophic forms [1,5,6,7,8]; heart rhythm changes, including atrioventricular conduction defects (AVCD); supraventricular tachycardia (SVT); atrial fibrillation (AF) and life threatening ventricular arrhythmias (VA). (However, other less frequently observed manifestations in children include structural cardiac defects such as valvular abnormalities) [1,7,8,9,10,11].

A detailed cardiac screening at the time of diagnosis and periodic surveillance may prevent major cardiovascular sequelae.

There is growing evidence that the detection and treatment prior to overt cardiac manifestations affords patients a comparably better outcome [12]. It is important to promote education regarding the importance of a proactive approach to screening, diagnosis, and management of cardiovascular complications of NMDs among clinicians. Multidisciplinary management is mandatory in this field, including for cardiologists, in genetic laboratory facilities, and for cardiogenetic teams, in order to offer appropriate timing for the diagnostic tool, potential therapeutics and expert neurologists [12,13].

In the following paragraphs we provide an updated “picture” of the selected NMDs and their related cardiac phenocopies, including potential therapeutic trials. 

The contribution of this study includes the possibility to revisit childhood manifestations from clinical cardiac involvement leading to novel therapeutic options of these disorders without incorporating these data within larger adult based cohorts and revisions. 

To the best of our knowledge, no similar updated revision on childhood cardiac manifestations and management options has been reported in the literature.

Figure 1 includes a summary of pediatric cardiac manifestations of the described NMDs in order of their frequency as reported in the literature.

## 2. Methods

We performed a review of previous studies documenting the co-existence of the selected NMDs and the cardiovascular involvement within the pediatric population. We searched PubMed for published studies without restriction on the date of publication and without restrictions on language, using the search terms “neuromuscular disorder”, “Friedreich’s Ataxia”, “FRDA”, “FA”, “myotonic dystrophy”, “DM1”, “DM”, “Kearns Sayre syndrome”, “KSS”, “*RYR1*”, “Laminopathy”, “*LMNA*” AND “heart”, “cardiovascular involvement”, “cardiac involvement”, “cardiomyopathy”, “arrhythmia”, “arrhythmic disorder” and “conduction defects”. We included review and cohort studies with the following specificity: NMD + cardiovascular disease + pediatric population. The papers were carefully read and reconsidered according to the above mentioned criteria. Two investigators performed the search independently. The references of the selected papers were crosschecked with the same inclusion condition. Duplicates were removed. Figure 2 includes the method used to conduct the literature search for the selected disorders. 

## 3. Results

In the following paragraphs we discuss the current knowledge related to the selected NMDs. Each paragraph includes a summary of the etiology, inheritance pattern, mechanism of action, principal clinical features, progressive cardiac manifestations (surveillance/management) and therapeutic options. Please note that Table 1 includes a summary of the main cardiac characteristics of children affected by the discussed NMDs. Moreover it includes the main management issues including follow up and treatment options. 

### 3.1. Friedreich’s Ataxia (FRDA) (Prevalence 1-9/100000, OMIM # 229300,601992) 

Friedrich’s ataxia (FRDA) is the most common spinocerebellar ataxia. The inheritance pattern is autosomal recessive. In the majority of patients, the underlying mechanism is a triplet repeat expansion of the GAA in the first intron of the gene encoding frataxin (*FXN*, also known as X25), which is located on chromosome 9. Few (2–5%) patients show different molecular mechanism that are unrelated to triplet repeat expansion. The normal GAA repeat size is ≤30 copies, and affected individuals typically have >70 triplets on each allele. If it is of an intermediate size (30–70) it is classified as a premutation, which is predisposed to expansion in future generations [12]. The exact role of frataxin in the development of FRDA is not known, and it is hypothesized that a deficiency of this protein causes cellular oxidative stress and iron storage in the heart, in a multifocal manner and, in some cases, the liver and spleen. Although frataxin deficiency is reported in several different tissues, its surplus in the heart is not yet well understood, but it may be related to mitochondrial dysfunction and the generation of oxidative stress in cardiomyocytes [14].

Clinical manifestations include ataxia of the limbs and trunk, a lack of deep tendon reflexes, sensory disturbances, skeletal deformities, diabetes, and cardiac involvement. High inter and intra familial variability has been reported. Frequently, neurological symptoms, which constitute the presenting sign, appear at the end of the first decade and they often (but not always) occur before cardiac symptoms [15]. 

The major cardiovascular signs/symptoms described in FRDA are CMP (57–81%) [16]. The age of onset ranges from the end of 1st decade to the end of 2nd decade of life. Further studies that include a cohort of children with FRDA are emerging and they identify a prevalent concentric hypertrophic cardiomyopathy (HCM) with both subclinical diastolic and systolic dysfunction [17]. HCM in FRDA is thought to represent around 8.6% of childhood onset HCM [5,6]. To the best of our knowledge, there are no multicentric studies indicating genotype–phenotype correlation. In fact, a recent study that included 68 children with FRDA confirmed this finding, but it also revealed an interesting result regarding more severe myocardial phenotype in adults carrying major triplet repeats (genotype–phenotype correlation). This study may reflect the idea of a progressive “expressivity” of the disease with age. In other words, in children with major triplet repeats an initiallack of genotype phenotype correlation might be seen. However, with progressive age this correlation does emerge (i.e., more severe phenotype with more triplet repeats) [18]. 

Child and colleagues hypothesized that two major myocardial patterns may occur, one being the “dystrophic” type with a prevalence of tissue fibrosis and the second being the “hypertrophic” type, as concentric or asymmetric myocardial hypertrophy is observed. Both conditions can lead to an end-stage heart failure [19].

Less frequent myocardial changes include asymmetrical septal hypertrophy, a left ventricular (LV) outflow gradient, and end-stage heart failure (HF) manifesting as dilated cardiomyopathy (DCM) [2,15]. The mean decrease in ejection fraction (EF) is age related but generally with maintains its normal values throughout childhood until the age of 22 years [20].

In children with FRDA, high risk arrhythmias are less frequently observed. In other words, most ECG changes are secondary to myocardial changes including QRST angles that are greater than 60° and 90°, an increased J-point on the first ECG, LV strain patterns and T-wave inversion [1,20]. Regarding the cardiac outcome in children, most probands reach adulthood without the need for heart transplantation (HT). Sudden cardiac death (SCD) is rarely observed [2,5,20]

According to the recommendations of the American Heart Association (AHA), in asymptomatic patients, electrocardiography and echocardiography are indicated at the moment of the FRDA diagnosis on an annual basis. Holter ECG surveillance can be performed every 1-4 years. In symptomatic patients, basic cardiac screening should be conducted more frequently and should be associated with Holter ECG monitoring [12]. In the late 1990s, coenzyme Q and Idebenone were identified as potential treatments for FRDA, and some studies have suggested their use in order to decelerate the progression of cardiac disease [20].

Other suggested therapeutic options include antioxidant molecules, iron-chelating agents and frataxin increase inducers (such as erythropoietin or specific histone deacetylase inhibitors or interferon).

While conventional drug therapy is extremely challenging, novel prospective treatments for FRDA are currently being developed, including pharmaceutical agents and gene therapy. The mechanism of function is mainly targeted on the base of the progressive damage caused by the triplet repeat increase in tissues. Molecules that improve mitochondrial function, such as Nrf2 activators, dPUFAs and catalytic antioxidants, as well as novel mechanism of frataxin augmentation and genetic modulation will eventually provide treatment for this serious disease [21].

### 3.2. Myotonic Dystrophy Type 1 (DM1) in Children (Congenital: Prevalence Very Rare; OMIM 160900)

Myotonic Dystrophy type 1 (DM1) is an autosomal dominant disease is majorly caused by a toxic CTG repeat expansion in the UTR of the *DMPK* located on chromosome 19q13.3. The longer the expansion, the earlier and more severe the phenotype [22]. In only 3–8% of DM1 do different variant repeats occur such as CCG and CGG, which are associated with late onset and a milder phenotype [23].

The *DMPK* CTG trinucleotide repeat length is mitotically unstable in individuals with DM1 which leads to somatic mosaicism in the size of the CTG expansion [23].

This variant is transcripted in a mutant RNA that disturbs the splicing of the CUG binding protein (CUG-BP) and the Muscle bind-like protein (MBNL). It then causes a phenomenon named “spliceopathy” caused by the sequestration of splicing factors, forming ribonuclear inclusions that disturbs cellular signaling and causes toxic effects on the muscle metabolism and RNA processing [24]. CUG-BP has been noted to bind to human cardiac troponin pre-mRNA, potentially explaining cardiac abnormalities in DM1 [10].

Normal alleles contain 5-34 CTG repeats, premutation alleles counts 35-49 CTG repeats and full-penetrance alleles which include >50 to >1000 [23]. The age of onset of DM1 is variable at the intrafamilial level due to the anticipation effect which leads to progressive severity along following generations, particularly when inheritance is maternal, leading to congenital (CDM) (0-1month)/childhood (ChDM)(1month to 18 years old) DM1 [22].

The exact global prevalence of CDM and ChDM forms is unknown but thought to represent 10–30% of overall DM1 levels [25]. In the near future, this percentage may decrease due to increasing methods of prenatal diagnosis since most of those affected by CDM1 and ChDM1 have an affected parent who will seek medical attention preconception. These conditions cause significant and early onset morbidity and mortality.

The clinical manifestations in adults include progressive distal muscle weakness, myotonia, early onset cataracts, progressive CMP, both brady and tachyarrhythmic events, gastrointestinal abnormalities and a Central Nervous System (CNS) dysfunction [22].

Most of the children are asymptomatic for dyspnea, palpitations or syncope [7,8,9,10,11].

In early onset forms, the cardiac phenotype might differ, and can include valvulopathies such as mitral valve prolapse [7,8,10], variable forms of CMP including HCM, left ventricular non compaction LVNC, and DCM [1,7,8].

Unlike in adults who present DCM, individuals affected by CDM can have HCM within the context of complex NMDs. The diagnosis of the underlying condition can be assisted by a positive family history of an affected parent with DM1 (personal observation, 2018).

Anomalies of the cardiac rhythm represent a major issue in CDM (16.7–25.8%) and ChDM(10%). Patients may have progressive conduction disturbances from first degree to complete atrioventricular (AV) block which can occur unexpectedly, leading to SCD. An HV interval > 70 msec can be predict the occurrence of a complete AV block and SCD. Both AF and ventricular tachyarrhythmias are observed in these patients which can predict SCD [26].

A cardiac screening program is a pivotal issue in determining risk stratification. It includes a yearly ECG, 24-h Holter monitoring, an echocardiogram and a stress test, and should be considered for CDM and ChDM [1,12].

According to the recommendations of the AHA for asymptomatic patients with DM, a cardiac evaluation should be indicated at diagnosis and on an annual basis using ECG and Holter ECG. An echocardiogram can be considered every 2 to 4 years. In symptomatic patients or in the occurrence of electrocardiographic changes, monitoring must be established at least annually and an invasive electrophysiological study might be considered for pacemakers (PMK) or Implantable Cardioverter Defibrillators (ICDs) implants [12]

An effective disease modifying treatment is not currently available. Novel therapies are emerging but remain unavailable on a practical level. A multidisciplinary management method is currently supported (regular surveillance and treatment of manifestations) [10].

Experimental novel therapeutic trials target different mechanisms aiming to prevent the toxic effect of CUG expansion. This be achieved using Antisense Oligonucleotides (AONs) to degrade the CUG expansion or binds to CUG expansion to inhibit RNA sequestration and sites for abnormal MBNL binding. The second mechanism can target the recombinant Adeno-associated viral (rAAV) which can stimulate an overexpression of MBNL1, in order to prevent sequestration. Emerging Clustered regularly interspaced short palindromic repeats (CRISPR/Cas) provide a third alternative mechanism which aims to cleave and degrade CUG mRNA expansion [24]. A fourth mechanism includes agents that increase muscle anabolism, such as testosterone, creatine, dehydroepiandrosterone, and recombinant insulin-like growth factor (IGF-1), and myostatin inhibitors.

These novel paths may be promising in the near future especially for ChDM who have not yet developed “permanent” or definite complications as a result of the disease.

### 3.3. Kearns Sayre Syndrome (KSS) (Prevalence 1-9/100000 OMIM #530000)

The Kearns-Sayre syndrome (KSS) is a rare heterogeneous neurodegenerative disorder that, in most cases, is caused by single, large heteroplasmic deletions of the mitochondrial DNA (mtDNA). The most common form is the deletion of 4.977bp, known as m.8470_13446del4977. Multiple pathogenic variants in either mtDNA or nuclear genes may also cause KSS [27]. It can be present in a variable percentage in different tissues including blood, urine, and muscle tissue.

KSS involves the musculoskeletal, central and peripheral nervous, endocrine and heart systems. It is characterized by a progressive external ophthalmoplegia, retinitis pigmentosa and an onset before the age of 20.

In addition to the abovementioned features, one of the following conditions must be present to establish the diagnosis: cerebellar ataxia, cerebrospinal fluid (CSF) proteins > 100 mg/dL and cardiac conduction disturbances [28]. 

Cardiac abnormalities are observed at multiple levels, including the structural (valvular and myocardial) and arrhythmic (both tachy- and bradyarrhythmias) levels.

Regarding structural abnormalities, in childhood onset KSS, rare multi-valvular involvement is described including mitral and tricuspid valve prolapse [29,30,31]. Other “peculiar” cardiac manifestation includes a single report from Chertkof in 2020 of a KSS diagnosed teenager presenting a dilated aortic root that seemed “idiopathic” [32].

CMPs are reported but not specified through a peculiar red flag phenocopy [29,33]. A heart transplant is rarely described in KSS [31,34]. Arrhythmic events can include: (1) progressive cardiac conduction defects up to complete AV bloc; (2) QT prolongation; (3) supraventricular tachyarrhythmias and (4) ventricular tachyarrhythmias. Due to the occurrence of these rhythm disturbances, syncope can occur in around half of the patients and SCD can occur in 20% [35,36].

Recently, our group [37] described the progressive conduction system involvement in KSS, revealing that the most common ECG anomalies are represented by an intraventricular and atrioventricular conduction delay which may occasionally occur as a first and dramatic clinical presentation. It is known that fascicular blocks in KSS carry a high risk of rapid, unpredictable and potentially fatal progression to complete atrioventricular block (cAVB). In fact, all of our pediatric onset studies of KSS have displayed intraventricular conduction anomalies, with similar conduction anomalies between children with classical and nonclassical deletions. Moreover, it was noticed that in most cases a left anterior fascicular block (LAFB) precedes the right bundle branch block (RBBB) causing a bi-fascicular block (Complete or advance atrioventricular block –cAVB or aAVB). This can evolve into a severe atrioventricular conduction defect.

Previous studies revealed an incidence of cAVB/aAVB in 40% of adults and children with a mean age of 13.8 ± 1.5 years [37,38].

The mechanism of the progressive cardiac conduction defect in KSS is unknown. However, it is thought that progressive clinical manifestations can be a result of progressive cardiomyocytes mtDNA depletion which leads to decreased mitochondrial enzyme activity [39,40].

The literature data report the median age of the first arrhythmic event as 15 years old [41].

In our cohort we observed tachyarrhythmic events both at the supraventricular level (focal atrial arrhthmias) 5/15 (33%) and for non-sustained ventricular tachycardia (NSVT) for 4/15 (26%). 

PMK implantation is often used in the pediatric population as a preventive treatment for major progressive conduction defects (in our cohort 11/15(73%), and a CD was implanted in 1 patient and an NSVT in 1 patient (6%). 

In the report by Di Mambro we underlined that the semestral screening program with 12-lead ECG and Holter ECG is crucial in preventing severe events in children and teenagers with KSS especially when seen in expert referral centers. This program is essential for the appropriate timing of the implantation of PMK/ICD [37]. Similarly, standardized echocardiogram monitoring every 6 months/1 year is also important for monitoring any structural cardiac/vascular changes [42].

Currently, there is no definitive treatment for KSS. Its management is mainly multidisciplinary and supportive in nature. Sporadic reports indicated that an integration with folic acid, coenzyme Q10 and antioxidants may be useful [42].

### 3.4. Ryanodine Receptor Type 1-Related Myopathies (RYR1-RM) (Pediatric Prevalence US 1:90.000, OMIM #11700, 255320)

Ryanodine receptor isoform 1 related myopathy (*RYR1*-RM) is the most frequently diagnosed congenital myopathy [Amburgey et al., 2011] [43]. This is a very heterogeneous group of disorders from a clinical/histological point of view [44].

The ryanodine receptor isoform 1 (*RYR1*) is the major Ca^2 ++^ channel expressed in skeletal muscle and is fundamental for the excitation-contraction process coupling. *RYR1* is present in the sarco membrane/endoplasmic reticulum and regulates the rapid intracellular release of Ca^2 +^ following the depolarization of the transverse tubules, and contributes to cellular Ca^2 +^ homeostasis in resting conditions [Zahradníková 1999] [45]. There are two other mammalian protein isoforms (*RYR2* and *RYR3*) which are expressed in multiple tissues and cell types as cardiac muscle (where *RYR2* is the predominant isoform) [46].

The mechanism underlying *RYR1*-RM may be related to the abnormal Ca2 + homeostasis and to correlated processes such as dysregulation of mitochondrial Ca2 uptake and consequent high oxidative and nitrative stress [47].

The inheritance mode of *RYR1*-RMs is both autosomal dominant and autosomal recessive, the latter is typically associated with more severe and early onset clinical phenotypes [48]. The age of onset varies from infancy, childhood and adulthood.

The classification of *RYR1* related abnormalities can be divided into two major categories according to a histological or a clinical classification.

On one hand, histological characterization (on muscle biopsy) *RYR1*-RM collects various nosological entities such as central core disease, multiminicore disease, core–rod myopathy, centronuclear myopathy, and a congenital fiber-type disproportion [49,50,51,52,53,54].

On the other hand, *RYR1* variants can determine different clinical phenotypes including interrelated conditions such as malignant hyperthermia susceptibility, exertional heat stroke, rhabdomyolysis-myalgia syndrome, King Denborough syndrome, and atypical periodic paralysis [55,56,57].

Some dominant *RYR1*-RM variants are allelic to the malignant hyperthermia susceptibility trait (MHS), which is a pharmacogenetics predisposition to malignant hyperthermia and severe adverse reactions to volatile anesthetics and muscle relaxants [58], and some of these imply an increased risk of MHS. For this reason, particular caution is recommended in anesthetic management in patients with deleterious variants in *RYR1* [59]. One of the included criteria for the clinical grading scale for malignant hyperthermia is cardiac involvement through unexplained sinus tachycardia, ventricular tachycardia or ventricular fibrillation (VF) [60].

RYR1 is greatly expressed in skeletal muscles and to a lesser extent in vascular smooth muscles but does not exert any effects on cardiomyocytes [61,62]. A potential mechanism of its involvement with the heart might be explained through indirect mechanisms related to secondary effects of RYR1 on metabolic cascades such as hyperthermia, and hyperkalemia.

Relatively few data provide summaries regarding the reasons for the specific cardiac involvement in *RYR1* [59,63]. However, both myocardial and arrhythmic events are reported mainly in the recessive form [59].

For the adult *RYR1*-RM population a report of HCM and conduction/arrhythmias was provided by Petri in 2019 [64]. To the best of our knowledge, myocardial involvement in children is not discussed. 

With regard to arrhythmias, the tachyarrhythmias that are occasionally reported are mainly ventricular which might be induced by sympathetic nervous system stimulation from hypercarbia, hyperkalemia, and catecholamine release. However observations of supraventricular tachyarrhythmias and conduction defects seem uncommon due to the lack of relevant reports [65]. An interesting point observed in the case reported by Hayakawa in 2019, of a little girl with a diagnosis of severe congenital *RYR1*-associated centronuclear myopathy caused by an inherited compound of heterozygous pathogenic/likely pathogenic rare variants in *RYR1*. At 5 months old, she developed both atrial tachycardia and sinus node dysfunction without CMP [66].

Although cardiac involvement appears infrequently in this nosological entity, regular follow-up is still important from the point of diagnosis.

International guidelines recommend, for NMDs (including *RYR1*-related), cardiac evaluation, ECG and an echocardiogram at the time of diagnosis and follow-up detecting the presence or development of any abnormal findings or cardiac symptoms [12]

Several attempts are being made to identify a therapy suited to the objective of treating congenital myopathies, specifically for those that are *RYR1* related.

In congenital myopathies (including related *RYR1*) a possible pharmacological treatment can include three main objectives: the direct modification of the altered function of the protein; better interactions between thin and thick filaments (for example, in some nemaline myopathies); and therapies aimed at nonspecifically ameliorating the downstream effects of the specific gene mutation [59]. 

An in vitro study that garnered good results included exon skipping to remove a pseudo-exon from the mRNA of a child with a recessive *RYR1*-related myopathy [67].

Other molecules that were described as potential therapeutic agents to be considered in *RYR1*-RM include *N*-acetylcysteine (reduce aberrant oxidative stress), Rycal (RYR1 closed channel stabilization), sodium 4 phenylbutyrate (chemical chaperone), 5-Aminoimidazole-4-carboxamide ribonucleoside (AICAR) and Dantrolene (RYR1 channel antagonist), Salbutamol/albuterol (potential enhancement of SERCA expression), and Carvedilol (Beta-blocker) [48,59].

### 3.5. Laminopathy (Prevalence Rare, LAMIN A/C OMIM-150330)

Laminopathies represent a heterogeneous group of rare clinical conditions caused by pathogenic variants in *LMNA* (OMIM 150330) located on chromosome 1q21.22 [68,69]. The gene has 5 isoforms (A, AD10, AD150, C and C2) encoded by alternative splicing. Laminae A and C are the two major isoforms, and the latter has the highest transcriptional expression [70,71].

They are widely expressed in the skeleton and the heart muscle but also in fat, blood vessels, skin and the nervous system [72].

An attempt at a clinical classification of these pathologies was made by Worman and Bonne [73] who divide the laminopathies into four main classes according to the main signs of presentation and symptoms, which are striated and cardiac muscle diseases, lipodystrophy syndromes, peripheral neuropathy and premature aging. There are at least 12 clinically distinct disorders that present disease-specific variants in *LMNA* [73].

Pathogenic variants in *LMNA* are related to 5–10% of the familial DCM and to 2–5% of the sporadic form (∼7% of all cases of idiopathic DCM) [74,75,76,77].

*LMNA*-related CMP is frequently associated with progressive arrhythmias such as tachyarrhythmias (supraventricular, and ventricular arrhythmias, the latter can lead to SCD) and progressive AV block indicating PMK implantation. 

Cardiac involvement in adult-onset laminopathies is well researched while literature is still underdeveloped regarding childhood-onset forms [3,78,79].

Laminopathies with cardiac involvement in children is a rare event that might be overlooked [3]. Arrhythmias seem to dominate the cardiac picture of these patients [80,81,82,83,84,85].

Structural cardiac involvement is reported mainly as CMPs such as restrictive cardiomyopathy (RCM), DCM and LVNC [81,83,84,86].

Recently, our group reported a cohort of six children with *LMNA* variants presenting a variable phenotype including congenital heart defects in 3/6: aortic coarctation (CoA), bicuspid aortic valve (BAV), mitral valve cleft, redundant mitral valve apparatus and ventricular septal defect (VSD). Two siblings presented a dilated ascending aorta, mildly dilated LV with mild hypertrabeculation of the lateral wall and a normally functioning but dilated left ventricle. Paroxysmal AF occurred in 3 patients (50% of cases) which is relatively high for such a rare manifestation in a pediatric age group which might represent a warning for this condition in children [3].

Trials are underway for this complex group of pathologies with the objective of identifying targeted therapies such as gene therapy. For example, the Hutchinson-Gilford Progeria Syndrome, a very complex laminopathy, appears to be an appropriate example for the development of gene therapy treatment due to the complex pathological mechanism which cannot be addressed efficiently using a single drug or multiple drug combinations due to the multiplicity of pathways involved. To this end, the objective is the development of larger animal models for a better understanding of the pathogenesis of these pathologies and the planning of targeted therapeutic strategies. This would be useful for all laminopathies and would expedite preclinical studies [87].

**Table 1 biomolecules-11-01578-t001:** Summary of the Main Cardiac Characteristics of Children Affected by the Discussed NMDs including Main Indications for Their Follow Up and Treatment Options.

NMD (OMIM)	Prevalence (in Children)	Gene	Mechanism	Inheritance	Structural HD	CMP	Heart Rhythm Changes	SCD	Cardiac Surveillace	Therapy
**FRDA(229300 601992)**	1–9/100000 (rare)	*FXN*	GAA-expansion in first intron (anticipation)	AR	-	Concentric HCM, DCM (End stage HCM)	ventricular arrhythmias	yes	**At diagnosis:** PE, ECG, echo, & 24 h ECG monitoring. **Asymptomatic:** at least annual screening. **Symptomatic:** consider more frequently. **Symptomatic+ arrhythmic events:** consider 24 h ECG monitoring and event recorder. Consider CMR.	Coenzyme Q, Idebenone, antioxidant and iron-chelating agents, frataxin increase inducers, HT in end-stage CMP.
**CDM and ChDM (#160900)**	Very rare, rare	*DMPK*	CTG- expansion in UTR (anticipation)	AD	valvulopathies	HCM, DCM, NCLV	PCD, AF, NSVT	yes	**At diagnosis:** PE, ECG, echo, & 24 h ECG monitoring, EST and SA-ECG. **Asymptomatic:****+ normal ECG and LVEF:** annual PE, ECG; 24 h ECG monitoring, EST and SA-ECG; every 2-4 years echo. **Symptomatic** **+ ECG anomalies:** at least annual screening (consider EPS for PM/ICD). Consider CMR.	Antiarrhythmic drugs, anticoagulant drugs, radiofrequency catheter ablation, HT in end-stage CMP. Several experimental trials in progress. Caution is recommended in anesthetic management.
**KSS (#** **530000)**	**1–9/100000**	m.8470_13446del4977	mtDNA deletion	matrilinear	MV or TV prolapse	DCM	PCD, long QT, NSVT, AF	yes	**At diagnosis:** PE, ECG, echo, & 24 h ECG monitoring, EST **Asymptomatic:** at least annual screening. **Symptomatic** **+ ECG anomalies:** consider more frequent screening (+EPS for PM/ICD). Consider CMR.	Folic acid, coenzyme Q10, antioxidants, HT.
* **RYR1** * **-RM(-)**	1:90.000 (rare)	*RYR1*	Missense/nonsense variants	AR, AD	-	-	PCD, tachyarrhythmias	-	**At diagnosis:** PE, ECG, echo. **Asymptomatic:** annual screening.	Dantrolene. Several experimental trials in progress. Caution is recommended in anesthetic management.
**L** **aminopathy**	**rare**	*LMNA*	Missense/nonsense variants	AR, AD	CoA, BAV, MV cleft, MVP, VSD	DCM, LVNC, RCM,	PCD, AF, SVT, NSVT	yes	**At diagnosis:** PE, ECG, echo, & 24 h ECG monitoring, EST. **Asymptomatic:** at least annual screening. **Symptomatic** **+** ECG anomalies: consider more frequent screening (+EPS for PM/ICD). Consider CMR.	Antiarrhymic drugs, synmptomatic therapy for HF, anticoagulant drugs. HT. Several experimental trials in progress. Caution is recommended in anesthetic management.

Abbreviations: AF, atrial fibrillation; AVB, atrioventricular block; BAV, bicuspid aortic valve; CAVB, complete atrioventricular block; CDM, congenital myotonic dystrophy; ChDM, childhood Myotonic Dystrophy; CMP, cardiomyopathy; CMR, cardiovascular magnetic resonance; CoAo, coarctation of the aorta; DCM, dilated cardiomyopathy; *DMPK*, Dystrophia Myotonica Protein Kinase gene; ECG, electrocardiogram; EPS, electrophysiology study; EST, exercise stress testing; FRDA, Friedreich’s Ataxia; HD, heart defect; HCM, hypertrophic cardiomyopathy; HF, heart failure; HT, heart transplantation; ICD, implantable cardioverter defibrillator; LVEF, left ventricular ejection fraction; LVNC, left ventricular noncompaction; MV cleft, mitral valve cleft; MVP, mitral valve prolapse; NMD, neuromuscular disorder; NSVT, non-sustained ventricular tachycardia; PCD, progressive conduction defect; PE, physical examination; PM, pacemaker; RCM, restrictive cardiomyopathy; VSD, ventricular septal defect; *RYR1*-RM, ryanodine receptor type 1-related myopathy; SA-ECG, signal average ECG; SCD, sudden cardiac death; SND, sinus node dysfunction; SVT, sustained ventricular tachycardia; TV, tricuspid valve; untranslated region; KSS, kearns sayre syndrome.

## 4. Conclusions

NMDs are a clinically and etiologically heterogeneous group of pathologies that are characterized by a multisystemic involvement. The cardiovascular system is one of the major causes of morbidity and of endpoint (mortality). The age of NMD’s onset is variable. In this review, we report on the importance of detecting the early onset rare photocopies cardiac manifestations in children including its structural (valvular), myocardial (CMPs) and arrhythmic (brady-tachyarrhythmias) aspects. The approach of these conditions must be multidisciplinary and be personalized in nature especially in the rare case of severe early onset forms. This can be crucial for the prognostic and the prospects for therapeutics. Due to a limited number of children with these NMDs, multicentric studies represent an avenue for the management and appropriate risk stratification of this specific sub-group.

## Figures and Tables

**Figure 1 biomolecules-11-01578-f001:**
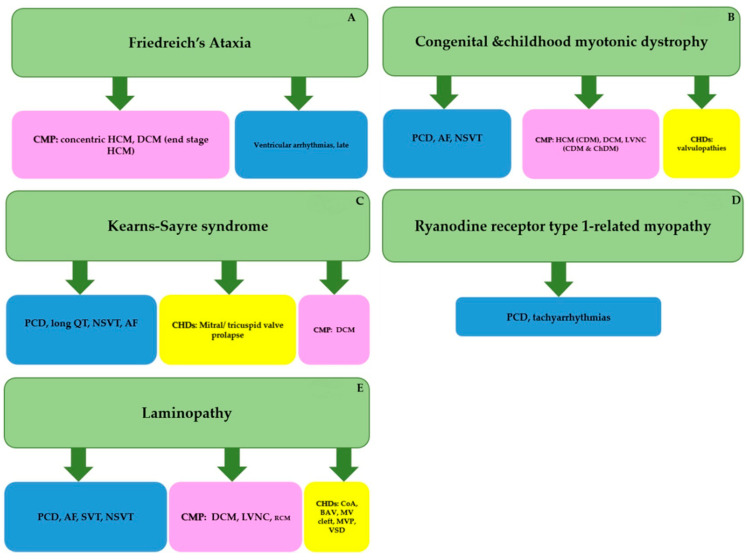
Pediatric cardiac manifestations of the described NMDs in order of their frequency as reported in literature: (**A**): Friedreich’s Ataxia (FRDA); (**B**): Congenital Myotonic Dystrophy (CDM) and Childhood Myotonic Dystrophy (ChDM); (**C**): Kearn Sayre Syndrome (KSS); (**D**): *RYR1*-RM, ryanodine receptor type 1-related myopathy; (**E**): Laminopathies. Abbreviations: AF, atrial fibrillation; AVB, atrioventricular block; BAV, bicuspid aortic valve; CHDs: congenital heart defects; CMP, cardiomyopathy; CoA, aortic coarctation; DCM, dilated cardiomyopathy; HCM, hypertrophic cardiomyopathy; LVNC, left ventricular non compaction; MV, mitral valve; MVP, mitral valve prolapse; NSVT, non-sustained ventricular tachycardia; PCD, progressive conduction defect; RCM, restrictive cardiomyopathy; SCD, sudden cardiac death; SND, sinus node dysfunction; SVT, supraventricular arrhythmias.

**Figure 2 biomolecules-11-01578-f002:**
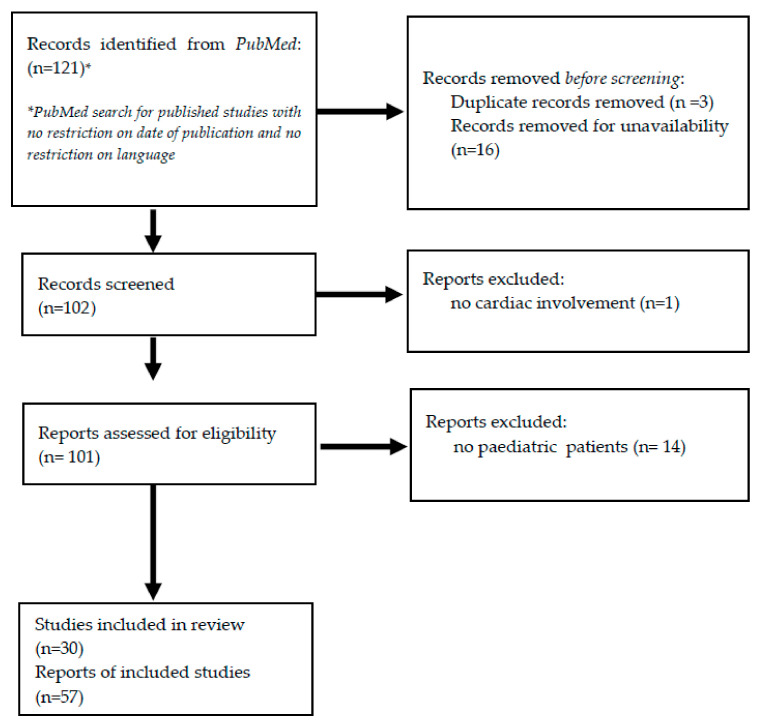
Method used to conduct the literature search for the selected disorders.

## Data Availability

The study did not report data that need to be reported.

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
