# Peer review of "Myocardial and Arrhythmic Spectrum of Neuromuscular Disorders in Children"

_biomolecules, 2021, doi:10.3390/biom11111578_

Round 1

Reviewer 1 Report

1) First and foremost, I think authors should systematically describe the purpose of your manuscript and the importance of your research topic.
2)
I don't think the research method is clearly presented in this manuscript.
3)
There are numerous typographical and grammatical errors throughout the manuscript.I strongly advise to get your manuscript proof-read by a native speaker before submission.

Please review the comments below for declining the manuscript:  

1) First of all, the manuscript ought to mention a type of review. Perhaps, it could be either a scoping review OR a narrative review. A mere review of the topic can be found searching various information via online.  

2) Most importantly, it does not have a methodology; for instance, there are no diagrams for search strategies. Many researchers use the PRISMA diagram to search terms; as you know, each database has different search strategies.  

3) It is rather ambiguous why this topic is interesting for the audience. I believe a clear statement is much needed to note why this research is vital at the end of the introduction section.  

4) Table 1 needs a better arrangement. On top of that, the main indication for their follow up is vague.   

5) Abbreviations should be part of the Figure legend.  

6) A major English proofreading is required.

Author Response

Query: 1) First and foremost, I think authors should systematically describe the purpose of your manuscript and the importance of your research topic.

Reply: We thank you for this comment. We have modified the paper in order to improve it to a scoping review. We agree with you that some information might be available online but for sure in a fragmented manner regarding this specific topic: a rarity within a rarity, in other words: children cardiac manifestations and management in specific NMDs. But the methodology and gathering of scientific data can give certain specificities especially for clinicians who are familiar with NMDs but less frequently familiar with either cardiac aspects or with children with NMDs. As a pediatric tertiary care research hospital  draining a population from center and south of Italy (around 25 millions inhabitants) we have learned not to take for granted that clinicians can have updated knowledge on children manifestations of rare diseases. Above all we learn everyday that children are not “small” grown-ups.

Query: 2) I don't think the research method is clearly presented in this manuscript.

Reply: We thank you for this observation. This query is repeated in number two where we answered accordingly but happy to explain more specifically if required.

Query: 3) There are numerous typographical and grammatical errors throughout the manuscript.I strongly advise to get your manuscript proof-read by a native speaker before submission.

Reply: We thank you for all the comments and observations. The paper was revised by native English speaker.

Please review the comments below for declining the manuscript: 

Query: 1) First of all, the manuscript ought to mention a type of review. Perhaps, it could be either a scoping review OR a narrative review. A mere review of the topic can be found searching various information via online.

Reply: We thank you for this comment. We have modified the paper in order to improve it to a scoping review. We agree with you that some information might be available online but for sure in a fragmented manner regarding this specific topic: a rarity within a rarity, in other words: children cardiac manifestations and management in specific NMDs. But the methodology and gathering of scientific data can give certain specificities especially for clinicians who are familiar with NMDs but less frequently familiar with either cardiac aspects or with children with NMDs. As a pediatric tertiary care research hospital  draining a population from center and south of Italy (around 25 millions inhabitants) we have learned not to take for granted that clinicians can have updated knowledge on children manifestations of rare diseases. Above all we learn everyday that children are not “small” grown-ups.

Query: 2) Most importantly, it does not have a methodology; for instance, there are no diagrams for search strategies. Many researchers use the PRISMA diagram to search terms; as you know, each database has different search strategies. 

Reply: We thank you for this observation. As requested we have added a specific diagram and a paragraph reporting the method of our review process.

Query: 3) It is rather ambiguous why this topic is interesting for the audience. I believe a clear statement is much needed to note why this research is vital at the end of the introduction section

Reply: We thank you for your observation which means that in the first version we did not communicate in a good way regarding  the main goal of the paper. We rephrased both abstract and introduction to give a better explanation.

The novelty of this study includes the possibility to revisit childhood manifestations of specific NMDs from basic, clinical, cardiac ending up with novel therapeutic trials of these disorders. Many of these data are available online but mainly incorporated within larger adulthood cohorts and revisions. To the best of our knowledge, no similar updated revision is reported in literature regarding  childhood cardiac manifestations and management options of these specific NMDs.

Query: 4) Table 1 needs a better arrangement. On top of that, the main indication for their follow up is vague.

Reply: We thank you for this observation. We had serious difficulty in uploading the figure. In this revision process we shall ask help from the Journal Manger since it is extremely difficult to upload tables and figures in incorporated manner.

Query: 5) Abbreviations should be part of the Figure legend. 

Reply:We thank you for your observation. We added the legend of abbreviations both to table and figures.

Query: 6) A major English proofreading is required.

Reply: We thank you for all the comments and observations. The paper was revised by native English speaker.

Reviewer 2 Report

In the manuscript ‘myocardial and arrhythmic Spectrum of Neuromuscular Disorders in Children’ the authors provide an overview of the most important neuromuscular disorders affecting childhood. They focus on the cardiac manifestations and describe some of the current possible management strategies.

This is an interesting review on the topic which still has the potential to be improved.

  1. One of my main concerns is that it’s not totally clear why the authors decided not to treat other relatively common NMD such as limb-girdle muscular dystrophy (LGMD1B is vaguely treated at the section of laminopathies), X-linked recessive muscular dystrophies (Duchenne, Becker and Emery-Dreifuss) or Barth syndrome.
  2. The structure of the text can be improved for example at the section on DM1: from page 5, line 226 the topics of the paragraphs are as follow
    • arrhythmic events
    • AVB
    • arrhythmic events
    • AVB/conduction disturbances for 2 paragraphs
    • Tachyarrhythmia
    • Pacemaker implantation
  3. When talking about arrhythmic events the terminology should be more clear. The authors should distinguish conduction disorder, supraventricular tachycardia and ventricular tachycardia. Syncope and sudden cardiac death are symptoms/consequences of the above mentioned (specially conduction disorders and ventricular events) and should not be described as ‘arrhythmias’.
  4. The authors do not always incorporate the guidelines of the American Heart Association 2017 (Feingold at al.) in their recommendation.
  5. There are multiple grammatical mistakes throughout the text. The text could benefit from a grammar check-up

Some minor issues

  1. Please include references in the introduction
  2. Figure 1 is very difficult to read. The authors mix ‘arrhythmia’ with ‘ECG changes’.
  3. Page 3, line 121. It’s not clear to me what the difference is between the two different cardiac patterns.
  4. When the authors refer to ‘parietal’ hypokinesia, I assume they refer to the lateral wall? Could you please clarify this.
  5. Page 4, line 178. It is unusual that Afib is associated with SCD. Could you please clarify this.
  6. Consider using the most common abbreviation of left ventricular non compaction: LVNC
  7. Page 8, line 359: supraventricular tachycardia rarely leads to SCD, please clarify this is the text.
  8. Table 1 is useful but difficult to read. The columns are too small. It’s sometime unclear what is the recommendation is for each situation because the text is too long. Consider summarizing some text: for example ‘palpitations, dizziness, syncope’= symptomatic or ‘PR>240ms, second or CAVB’= any AVB

Author Response

Query: 1. One of my main concerns is that it’s not totally clear why the authors decided not to treat other relatively common NMD such as limb-girdle muscular dystrophy (LGMD1B is vaguely treated at the section of laminopathies), X-linked recessive muscular dystrophies (Duchenne, Becker and Emery-Dreifuss) or Barth syndrome

Reply: We thank you for revising our manuscript. We appreciate all the work done. We completely agree with you regarding the importance of DMD, BMD. We opted not to discuss them due to extensive series regarding cardiac aspects in children and due to words limit that we already had overcome in the current review. Regarding Barth Syndrome we did not include it since we thought it might be considered a more metabolic / mitochondrial condition rather than NMD. However, we are happy and glad to receive your feedback and open to include them if you think it is mandatory for the review.

Query: 2. The structure of the text can be improved for example at the section on DM1: from page 5, line 226 the topics of the paragraphs are as follow

arrhythmic events

AVB

arrhythmic events

AVB/conduction disturbances for 2 paragraphs

Tachyarrhythmia

Pacemaker implantation

Reply: we thanks you for your observation. We did the requested changes.

Query: 3.  When talking about arrhythmic events the terminology should be more clear. The authors should distinguish conduction disorder, supraventricular tachycardia and ventricular tachycardia. Syncope and sudden cardiac death are symptoms/consequences of the above mentioned (specially conduction disorders and ventricular events) and should not be described as ‘arrhythmias’.

Reply: we apologize for this. We have rephrased the terms throughout the manuscript.

Query: 4.  The authors do not always incorporate the guidelines of the American Heart Association 2017 (Feingold at al.) in their recommendation.

Reply: Thank you for your comment, wherever missing, we included the guidelines of the American Heart Association 2017.

Query: 5.  There are multiple grammatical mistakes throughout the text. The text could benefit from a grammar check-up

Reply: We thank you for all the comments and observations. The paper was revised by native English speaker.

Some minor issues

Query: 1. Please include references in the introduction

Reply:  We apologize for this.  References are now included in the introduction

Query: 2.  Figure 1 is very difficult to read. The authors mix ‘arrhythmia’ with ‘ECG changes’

Reply: We thank you for this observation. We had serious difficulty in uploading the figure. In this revision process we shall ask help from the Journal Manger since it is extremely difficult to upload tables and figures in incorporated manner.

Query: 3.  Page 3, line 121. It’s not clear to me what the difference is between the two different cardiac patterns.

Reply: We thank you for your comment. We rephrased the sentence. We agree with you. It was not well written.

Query: 4.  When the authors refer to ‘parietal’ hypokinesia, I assume they refer to the lateral wall? Could you please clarify this.

Reply: We thank you for your comment. We rephrased the sentence. We agree with you. It was not well written.

Query: 5. Page 4, line 178. It is unusual that Afib is associated with SCD. Could you please clarify this.

Reply: We thank you for this comment. According to Groh et al., atrial tachyarrhythmia is the only independent predictor of both sudden death and death from progressive NM respiratory failure [Groh WJ, Groh MR, Saha C et al., NEJM 2008]. We hope that we answered correctly the query and happy to receive your precious comments.

Query: 6. Consider using the most common abbreviation of left ventricular non compaction: LVNC

Reply: we thank you for this observation. We have standardized the abbreviation

Query: 7. Page 8, line 359: supraventricular tachycardia rarely leads to SCD, please clarify this is the text.

Reply: Many thanks for your comment. We made it clearer since the “,” was not please correctly. We rephrased the sentence and hope it is fine.

Query: 8. Table 1 is useful but difficult to read. The columns are too small. It’s sometime unclear what is the recommendation is for each situation because the text is too long. Consider summarizing some text: for example ‘palpitations, dizziness, syncope’= symptomatic or ‘PR>240ms, second or CAVB’= any AVB

Reply: We thank you for this observation. We had a real difficulty in uploading the table. In this revision process we shall ask help from the Journal Manger since it is extremely difficult to upload the table in incorporated manner. We standardized the applied parameters.

Round 2

Reviewer 1 Report

Please review the below:

1) As stated in the comments "To the best of our knowledge, no similar updated revision is reported in literature regarding  childhood cardiac manifestations and management options of these specific NMDs."

It is vital to note the purpose of the update and mention the previous results. More importantly, it ought to include how long has been since the first update. i.e) : A 5-year update

There are numerous SRs that have updated MNs in 5 or 10 year period.

2) If this MN were to be a Scoping review, authors should review the framework. Please review: https://www.tandfonline.com/doi/abs/10.1080/1364557032000119616

3) Keyword search strategies per database must be included as a supplement. i.e) Pubmed using Mesh terms and titles/abstracts, etc.

Reviewer 2 Report

I would like to thank the authors for for their detailed review of the text.

The manuscript, figures and tables have been significantly improved.